# Gradient Regularized $V$-Learning for Dynamic Treatment Regimes

**Yao Zhang**
University of Cambridge
yz555@cam.ac.uk

**Mihaela van der Schaar**
University of Cambridge
University of California, Los Angeles
The Alan Turing Institute
mv472@cam.ac.uk

## Abstract

Deciding how to optimally treat a patient, including how to select treatments over time among the multiple available treatments, represents one of the most important issues that need to be addressed in medicine today. A dynamic treatment regime (DTR) is a sequence of treatment rules indicating how to individualize treatments for a patient based on the previously assigned treatments and the evolving covariate history. However, DTR evaluation and learning based on offline data remain challenging problems due to the bias introduced by time-varying confounders that affect treatment assignment over time; this may lead to suboptimal treatment rules being used in practice. In this paper, we introduce Gradient Regularized $V$-learning (GRV), a novel method for estimating the value function of a DTR. GRV regularizes the underlying outcome and propensity score models with respect to the optimality condition in semiparametric estimation theory. On the basis of this design, we construct estimators that are efficient and stable in finite samples regime. Using multiple simulation studies and one real-world medical dataset, we demonstrate that our method is superior in DTR evaluation and learning, thereby providing improved treatment options over time for patients.

## 1 Introduction

Clinical decision-makers regularly face the daunting challenge of choosing from multiple treatment options and treatment timings. While clinical trials represent the gold standard for causal inference, clinical trials for longitudinal studies are expensive to conduct. They have few patients and narrow inclusion criteria, and usually do not follow patients for long periods of time. Leveraging increasingly available observational data about patients, such as electronic health records, represents a more viable alternative for developing individualized treatment plans over time. Across multiple communities, including machine learning, statistics and economics, increasing attention has been paid to the need to understand how decision-makers decide which treatments to give patients, and how to construct treatment rules based on their static information [1, 2, 3, 4, 5, 6, 7].

Treatment individualization and adaptation over time are crucial for managing chronic diseases. For example, the time-varying patient information, such as side-effect severity and treatment adherence, drives the treatment for major depressive disorder [8]; clinicians routinely adjust therapy based on the risk of toxicity and antibiotics resistance in treating cystic fibrosis [9]. Unlike in static settings, developing time-varying treatment rules in longitudinal settings poses unique opportunities to understand how diseases evolve under different treatment plans, how individual patients respond to medication over time, and which timings are optimal for assigning treatments. Dynamic treatment regimes (DTRs) [10, 11] offer an attractive causal inference framework that serves this purpose. A DTR is a sequence of time-varying treatment rules that determine which treatment to provide

at each time step, given the patient's ongoing observation and evolving treatment history. These time-varying treatment rules are also known as adaptive treatment strategies [12, 13, 14, 15, 16] or treatment policies [17, 18, 19]. DTRs provide an effective vehicle for several application areas that have recently been discussed in biomedical literature, including personalized management of chronic conditions for cancer, diabetes, and mental illnesses [20]. Moreover, DTRs can be used to determine when to use chemotherapy or radiotherapy on lung cancer patients for controlling the growth of their tumour volume, when to stop using antibiotics treatment to decrease the white blood cell count (indicative of severe illness and poor outcome) for ICU patients, and when to use ventilation on patients with COVID-19 to maximize their survival outcomes.

To estimate the value function of a DTR, there are three classes of estimators to consider in the literature. Each of these classes has limitations in practice. It is well known that under some circumstances importance sampling-based estimators have high variance, while regression-based estimators have large biases. Doubly robust (DR) estimators have desirable asymptotic properties, but they can still suffer from high variance in finite samples, due to the unstable inverse propensity score (IPS) product. More in-depth discussion of existing methods is provided in Section 3.

In this paper, we introduce Gradient Regularized $V$-learning (GRV) as a new method for estimating the value function of a DTR. The GRV estimator is constructed to have stable finite sample behaviour while achieving the asymptotic efficiency. GRV achieves this by regularizing the underlying machine learning models to satisfy the estimating equation in semiparametric estimation theory. We prove theoretically that the estimating equation is satisfied when our proposed regularizer is minimized. Our work can be viewed as an extension of the targeted regularization (TR) method [21] to DTR settings. TR is a debiased method for estimating the average treatment effect (ATE) in static settings. Our regularizer is designed to solve the complex estimating equation for DTR evaluation, which involves the outcome and propensity score models across different time steps. Our regularizer can also be applied to estimate the average long-term effect of multiple treatments in longitudinal studies. With experiments on multiple simulation studies, we demonstrate that our method is superior to baseline methods in terms of accuracy in estimating value functions and learning better DTRs. We also demonstrate the application of our method on a real-world dataset extracted from the Medical Information Mart for Intensive Care (MIMIC) database [22].

## 2 DTR Problem

Under the standard assumptions of sequential ignorability, consistency and overlap [10, 23, 24] in the potential outcome framework for causal inference, we consider a dataset consisting of treatment trajectories from a population of $N$ units. For each $i \in [N] = \{1, \ldots, N\}$, unit $i$ has a baseline covariate vector in $Z_i \in \mathcal{Z}$. At each time $t \in [T] = \{1, \ldots, T\}$, unit $i$ has a covariate vector $X_{i,t} \in \mathcal{X}_t$, outcome variable $Y_{i,t} \in \mathcal{Y}_t$, and a one-hot treatment vector $A_{i,t} \in \mathcal{A} = \{0,1\}^K$ which indicates which of the $K$ treatment options is assigned to the unit $i$. In the case of two treatments, we have four options: no treatments, treatment 1, treatment 2, and both treatments.

We collect all the observations about unit $i$ by $O_i = (Z_i, X_{i,[T]}, A_{i,[T]}, Y_{i,[T]})$. The vectors $O_{[N]}$ are $N$ i.i.d sampled copies of $O \in (Z, X_{[T]}, A_{[T]}, Y_{[T]}) \sim P(O) = P$. We denote the history up to time $t$ by $H_t = (X_{[t]}, A_{[t]}, Z) \in \mathcal{H}_t$, $\tilde{H}_t = (X_{[t]}, A_{[t-1]}, Z) \in \tilde{\mathcal{H}}_t$ and the baseline covariates $Z$ by $H_0$. Then we can factorize the joint distribution $P$ as

$$P(O) = P(H_0) \prod_{t=1}^{T} P(X_t \mid H_{t-1}) P(A_t \mid \tilde{H}_t) P(Y_t \mid H_t) = g(O) f(O) h(O), \qquad (1)$$

where $g(O) = \prod_{t=1}^{T} P(A_t \mid \tilde{H}_t)$ is the intervention distribution, $f(O) = \prod_{t=1}^{T} P(Y_t \mid H_t)$ is the outcome distribution, and $h(O) = P(H_0) \prod_{t=1}^{T} P(X_t \mid H_{t-1})$ is the covariate distribution.

We denote the time $t$ treatment rule by $d_t : \tilde{\mathcal{H}}_t \to \mathcal{A}$, which assigns the treatment $A_t$ as a function of the history $\tilde{H}_t$. A dynamical treatment regime (DTR), $d := d_{1:T} = \{d_1, \ldots, d_T\}$, is a set that collects all the treatment rules over time. We let $P_{d_{t:T}}$ denote the joint distribution under the intervention of $d_{t:T}$, i.e., replacing $P(A_s \mid \tilde{H}_s)$ in $P(O)$ with the indicator function $\mathbb{1}(A_s = d_s(\tilde{H}_s))$, for

$s = t, \ldots, T$. The time $t$ value function under $d_{t:T}$ is given as

$$V_t(d_{t:T}) = \mathbb{E}_{P_{d_{t:T}}} \left[ \sum_{r=t}^{T} Y_r \right] = \int \sum_{r=t}^{T} Y_r \frac{dP_{d_{t:T}}}{dP} dP = \mathbb{E}_P \left[ \frac{\prod_{s=t}^{T} \mathbb{1} \left( A_s = d_s(\tilde{H}_s) \right)}{\prod_{s=t}^{T} P(A_s | \tilde{H}_s)} \sum_{r=t}^{T} Y_r \right]. \quad (2)$$

The optimal DTR is given as the optimizer $d^* = \arg\max_d V(d)$ where $V(d)$ is the value function over the entire treatment trajectory. To learn the optimal DTR $d^*$, one can first consider developing a sample-efficient and low variance estimator to evaluate $V(d)$, as will be discussed in Section 3.

## 3  Related works

In the literature, DTR learning in causal inference and off policy evaluation (OPE) in batch reinforcement learning (BRL) are the two branches of methods that attempt to solve the estimation problem of $V(d)$. Here, we summarize the existing value function estimators in three classes.

**Importance Sampling estimators.** The importance sampling (IS) estimator of $V_t(d_{t:T})$ is given by the empirical version of the R.H.S of Equation (2), with $P(A_s \mid \tilde{H}_s)$ in the denominator approximated by a propensity score model $\hat{g}_s(A_s, \tilde{H}_s)$. In the DTR literature, backward outcome weighted learning (BOWL) [6] is a method that derives the treatment rule $\hat{d}_t$ by optimizing the IS estimator of $V_t(d_t, \hat{d}_{t+1:T})$ backwardly through time given the previously optimized rules $\hat{d}_{t+1:T}$. Simultaneous outcome weighted learning (SOWL) [6] optimizes the treatment rules in $d$ jointly based on the IS estimator of $V(d)$. In BRL, IS estimators [25, 26, 27] are used to evaluate the value function under a target policy by reweighting the rewards in the historical data with the probability ratio of the target policy and the policy that generates the data. IS estimators are known to be consistent and unbiased but suffer high variance due to the inverse propensity score product.

**Regression-based estimators.** Batch $Q$-learning (BQL) [28] is a classic machine learning method that uses regression models to directly estimate the value function of a DTR. The term "batch" is used to emphasize an important difference from standard $Q$-learning [29, 30], in which learning occurs only after the collection of the training set. Alternatively, $A$-Learning [10, 31] models regret functions which measure the loss incurred by not following the optimal DTR. More discussion on the relationship between $Q$- and $A$-learning can be found in [32]. The variants of BQL in OPE are known as directed methods (DM) [33, 34, 35]. Here, we briefly describe BQL and its limitations.

In BQL, we define the time $T$ $Q$-function as $Q_T(a_T, \tilde{h}_T) = \mathbb{E}[Y_T \mid A_T = a_T, \tilde{H}_T = \tilde{h}_T]$. When the underlying data generating distribution $P$ is known, dynamic programming (DP) [36, 37] shows that $d_T^*(h_T) \in \arg\max_{a_T \in \mathcal{A}} Q_T(a_T, \tilde{h}_T)$ and recursively $d_t^*(h_t) \in \arg\max_{a_t \in \mathcal{A}} Q_t(a_t, \tilde{h}_t)$ where $Q_t(a_t, \tilde{h}_t) = \mathbb{E}[Y_t(a_t) + \max_{a_{t+1} \in \mathcal{A}} Q_{t+1}(a_{t+1}, \tilde{H}_{t+1}) \mid A_t = a_t, \tilde{H}_t = \tilde{h}_t]$. BQL approximates each $Q_t$ using some machine learning models $\hat{Q}_t$, and updates them backwards through time $t = T, T-1, \ldots, 1$ by solving the optimization problem:

$$\hat{Q}_t \in \arg\min_{Q_t \in \mathcal{F}} \frac{1}{N} \sum_{i=1}^{N} \left[ Y_{i,t} + \max_{a_{t+1} \in \mathcal{A}} \hat{Q}_{t+1}(a_{t+1}, \tilde{H}_{i,t+1}) - Q_t(A_{i,t}, \tilde{H}_{i,t}) \right]^2. \quad (3)$$

The optimal time $t$ treatment rule is derived as $\hat{d}_t(\tilde{H}_t) = \arg\max_{a_t \in \mathcal{A}} \hat{Q}_t(a_t, \tilde{H}_t)$. Despite BQL being able to produce low-variance estimators, the bias in these estimators is hard to quantify. Learning the $Q$-functions with Equation (3) implicitly restricts the solution to be in a postulated function class $\tilde{\mathcal{F}}$ and induces a corresponding class of DTRs given by the arg max of the postulated $Q$-functions. The work [38] shows that even when the optimal DTR resides in the class of induced DTRs, $Q$-learning can still fail to be consistent when $\tilde{\mathcal{F}}$ is misspecified. Hence, in $Q$-learning, the quality of DTR optimization will rely on the correctness of postulated regression models which we can not quantify easily. This limitation of $Q$-learning shows the advantage of simultaneous learning of $V(d)$ in some importance sampling based methods.

**Doubly robust estimators.** Developing doubly robust (DR) estimators is a well-studied extension in the literature [39, 40, 41, 42, 43] to improve value function estimation in terms of sample efficiency and robustness to model misspecification. These DR estimators have a similar expression as the well-known augmented inverse probability weighted (AIPW) estimator [44, 45] for the average

treatment effect (ATE) estimation in static settings. Despite their asymptotic optimality, they can be unstable in finite samples due to the IPS product. For example, the AIPW estimator in [42] is given as

$$\hat{V}_{\text{AIPW}}(d) = \frac{1}{N} \sum_{i=1}^{N} \left\{ \frac{C_\delta}{\hat{g}_{i,1:T}} \Big( \sum_{t=1}^{T} Y_{i,t} \Big) + \sum_{t=1}^{T} \frac{C_{\delta,t}}{\hat{g}_{i,1:t}} \hat{Q}_t(A_{i,t}, \tilde{H}_{i,t}) \right\}, \qquad (4)$$

where $C_\delta$ and $C_{\delta,t}$ are some indicator functions of whether DTR treatments align with observed data, and $\hat{g}_{i,1:t}$ is a function of $\prod_{s=1}^{t} \hat{g}_t(A_{i,s}, \tilde{H}_{i,s})$. By adaptively using a DR estimator for the value function in part of a trajectory and then using a regression-based estimator for the remainder, the weighted doubly-robust (WDR) and MAGIC estimator in [40] offer more accurate evaluation than the standard DR estimators. This result sheds the light on the need for an estimator that has a stable expression against high variance due to the IPS products while achieving the asymptotic optimality. Next, we introduce our method, Gradient Regularized $V$-Learning (GRV), that fills this need.

## 4    Gradient Regularized $V$-Learning

This section is organized as follows: (1) we first introduce a set of the outcome and propensity models, $\mathcal{M}_{t:T}$, required for estimating the value function $V_t(d_{t:T})$; (2) we discuss under what condition $\hat{\mathcal{M}}_{t:T}$ can construct an efficient estimator of $V_t(d_{t:T})$ in semiparametric theory; (3) we describe our neural network architecture that parameterizes the models in $\hat{\mathcal{M}}_{t:T}$, and a theory that demonstrates that our proposed regularizer can encourage the models to satisfy the optimality condition of efficient estimators; and (4) we briefly discuss the two GRV based DTR learning algorithms.

### 4.1    Set up

The time $t$ value function $V_t(d_{t:T}) = \mathbb{E}_{P_{d_{t:T}}} \big[ \sum_{r=t}^{T} Y_r \big]$ is defined as the sum of the mean outcomes after time $t-1$ under the intervention of $d_{t:T}$, in which

$$Y_r = Y_r(d_{t:r}) = \sum_{a_{t:r} \in \mathcal{A}^{r-t}} \prod_{s=t}^{r} \mathbb{1}(d_s(\tilde{H}_s) = a_s) Y_r(a_{t:r}),$$

for $r = t, \ldots, T$. To model these intervened outcomes using the data generated by the distribution $P(O)$ defined in Equation (1), we make no parametric assumptions on the joint covariate distribution $h(O)$, but make parametric assumptions on the joint outcome distribution $f(O)$ and intervention distribution $g(O)$. We let $\mu_{t,t}(A_t, \tilde{H}_t)$ denote the mean of $Y_t$ under the distribution $P(Y_t \mid H_t)$, and $g_t(A_t, \tilde{H}_t)$ denote the propensity score $P(A_t \mid \tilde{H}_t)$. We let $\mu_{t,r}(d_{t:r}, \tilde{H}_t)$ denote the mean of $Y_r$ given the intervention of $d_{t:r}$ and the history $\tilde{H}_t$. In our notation of $\mu_{t,r}$, "$t$" indicates the given history is up to time $t$ and "$r$" is the time of the future outcome $Y_r$. This long-term expectation can be computed via sequential regression. Starting backward from time $r$, given $\mu_{r,r}(A_r, \tilde{H}_r)$, we can compute $\mu_{r,r}(d_r, \tilde{H}_r)$ as $\mu_{r,r}(d_r(\tilde{H}_r), \tilde{H}_r) = \mathbb{E}\big[ Y_r \mid A_r = d_r(\tilde{H}_r), \tilde{H}_r \big]$. Then by the law of iterated expectations, for $s = t, \ldots, r-1$, we can compute

$$\mu_{s,r}(d_{s+1:r}, H_s) = \mathbb{E}\big[ \mu_{s+1,r}(d_{s+1:r}, \tilde{H}_{s+1}) \mid H_s \big]. \qquad (5)$$

In practice, $\mu_{s,r}(d_{s+1:r}, H_s)$ is computed by regressing $\mu_{s+1,r}(d_{s+1:r}, \tilde{H}_{s+1})$ onto $H_s$. After $r-t$ regressions, we obtain the wanted expectation $\mu_{t,r}(d_{t:r}, \tilde{H}_t)$. We let $\mathcal{M}_{t:T}$ collect all the mean outcome and propensity score models that we need to compute in this sequential procedure for estimating the time $t$ value function $V_t(d_{t:T})$. It is worth noting that the simplest treatment rules are the ones that assign the same treatment to all the units. The method we propose later also works for estimating the average long-term outcome $\mathbb{E}\big[ Y_r(a_{t:r}) \big]$ given some specific treatment plan $A_{t:r} = a_{t:r}$. We can estimate the effect of a target treatment plan by contrasting its outcome $Y_r$ with the one under some simple treatment plans, such as the plan that never assigns any treatments.

### 4.2    Efficiency

We now turn to discuss how to construct a consistent and efficient estimator of $V_t(d_{t:T})$ based on semiparametric theory. For a complete textbook presentation of the theory, we refer the reader to

[46, 47, 48, 49, 50]. We let $\mathcal{P}_{t:T}$ denote the space of the data distribution $P_{t:T}$ of $\tilde{H}_t$, $X_{t+1:T}$, $A_{t:T}$ and $Y_{t:T}$, and $\hat{P}_{t:T}$ denote a estimate of $P_{t:T}$ with a set of nuisance models $\hat{\mathcal{M}}_{t:T}$. We define the target parameter mapping as $\Psi_t : \mathcal{P}_{t:T} \to \mathbb{R}$ such that $\Psi_t(P_{t:T}; d_{t:T}) = V_t(d_{t:T})$.

For parametric models with finite-dimensional parameters, we can define the nuisance tangent space as the subspace spanned by the nuisance score vector which is the gradient of $\log P_{t:T}$ w.r.t the parameters of the nuisance models. Similarly, we can define the target score vector for the target parameters. The efficient influence curve (EIC) is a scaled residual obtained by projecting the target score vector onto the nuisance tangent space. The EIC is the influence curve with the smallest variance equal to the inverse of the Fisher Information, which is well-known to be the Cramer-Rao lower bound on the variance of unbiased estimators of the target parameters and attainable via maximum likelihood estimation (MLE) under regularity assumptions.

In semiparametric models, we cannot define scores for the infinite-dimensional nuisance parameters. However, if we assume that the target parameter mapping $\Psi_t$ is a pathwise differentiable functional at each distribution in $\mathcal{P}_{t:T}$, the EIC is given as the canonical gradient of $\Psi_t$, $D_t^*(\hat{P}_{t:T}; d_{t:T})$. This defines an estimator that is asymptotically efficient and linear with the EIC. To see this, we use the fact that the pathwise differentiable $\Psi_t(\hat{P}_{t:T}; d_{t:T}) = \hat{V}_t(d_{t:T})$ admits a von Mises Expansion [47]:

$$\hat{V}_t(d_{t:T}) - V_t(d_{t:T}) = (\hat{\mathbb{P}}_{t:T} - \mathbb{P}_{t:T})D_t^*(\hat{P}_{t:T}; d_{t:T}) + R_2(\hat{P}_{t:T}, P_{t:T}),$$

where $R_2(\hat{P}_{t:T}, P_{t:T})$ is the second-order remainder. Using the fact that EIC has zero mean such that $\hat{\mathbb{P}}_{t:T} D_t^*(\hat{P}_{t:T}; d_{t:T}) = 0$, we can decompose the first-order bias $-\mathbb{P}_{t:T} D_t^*(\hat{P}_{t:T}; d_{t:T})$ as

$$(\mathbb{P}_N - \mathbb{P}_{t:T})\big[D_t^*(\hat{P}_{t:T}; d_{t:T}) - D_t^*(P_{t:T}; d_{t:T})\big] + (\mathbb{P}_N - \mathbb{P}_{t:T})D_t^*(P_{t:T}; d_{t:T}) - \mathbb{P}_N D_t^*(\hat{P}_{t:T}; d_{t:T}).$$

In the decomposition, the first term will be $O_{P_{t:T}}(1/\sqrt{N})$ under empirical process conditions, e.g., if the nuisance models in $\hat{\mathcal{M}}_{t:T}$ are regular enough so that $D_t^*(\hat{P}_{t:T}; d_{t:T})$ lies in a Donsker class of functions, or if sample splitting is used so that the nuisance models are constructed on separate data. The second term is a sample-average of a fixed function $D_t^*(P_{t:T}; d_{t:T})$, and thus converges to a normal distribution after $\sqrt{N}$-scaling, by the central limit theorem. The remainder $R_2(\hat{P}_{t:T}, P_{t:T})$ can be $O_{P_{t:T}}(1/\sqrt{N})$ even when nuisance models converge at slower rates. Overall, we can have

$$\hat{V}_t(d_{t:T}) - V_t(d_{t:T}) = -\mathbb{P}_N D_t^*(\hat{P}_{t:T}; d_{t:T}) + O_{P_{t:T}}(1/\sqrt{N}), \qquad (6)$$

which implies that $\hat{V}_t(d_{t:T})$ is a $\sqrt{N}$-consistent estimator of $V_t(d_{t:T})$ if we can set the sample-average EIC $\mathbb{P}_N D_t^*(\hat{P}_{t:T}; d_{t:T})$ to zero. The variance of the EIC gives the generalized Cramer-Rao lower bound of the estimator variance [50]. The estimator $\hat{V}_t(d_{t:T})$ reaches this lower bound asymptotically and hence efficient. A general introduction of deriving the EIC for large models can be found in [51]. In our case, the EIC takes the form

$$D_t^*(P_{t:T}; d_{t:T}) = \sum_{r=t}^{T} \Big(\mathbb{E}\big[Y_r(d_{t:r}) \mid \tilde{H}_t\big] - \mathbb{E}\big[Y_r(d_{t:r})\big]\Big) + \sum_{r=t}^{T}\sum_{s=t}^{r} D_{s,r}^*(P_{t:T}; d_{t:T}). \qquad (7)$$

Let $\delta_{t:k}/g_{t:k}$ denote the IPS product $\prod_{m=t}^{k} \mathbb{1}(A_m = d_m(\tilde{H}_m))/\prod_{m=t}^{k} P(A_m \mid \tilde{H}_m)$. In the second term of Equation (7), as $s = t, \ldots, r-1$,

$$D_{s,r}^*(P_{t:T}; d_{t:T}) = \frac{\delta_{t:s}}{g_{t:s}}\Big(\mathbb{E}\big[Y_r(d_{s+1:r}) \mid \tilde{H}_{s+1}\big] - \mathbb{E}\big[Y_r(d_{s:r}) \mid \tilde{H}_s\big]\Big), \qquad (8)$$

and as $s = r$,

$$D_{r,r}^*(P_{t:T}; d_{t:T}) = \frac{\delta_{t:r}}{g_{t:r}}\Big(Y_r - \mathbb{E}\big[Y_r(d_r) \mid \tilde{H}_r\big]\Big). \qquad (9)$$

Suppose we define a fluctuated estimate $\hat{P}_{t:T,\epsilon}$ of $P_{t:T}$ so that a small change in $\epsilon$ corresponds with a maximal small change in the estimator $\Psi_t(\hat{P}_{t:T,\epsilon}; d_{t:T})$ on the data, the local maximum $\epsilon^*$ is given by solving the estimating equation: $\mathbb{P}_N D_t^*(\hat{P}_{t:T,\epsilon*}; d_{t:T}) \approx 0$. Targeted minimum loss estimation (TMLE) [52, 53, 54, 55] is an estimation procedure that relies on an extra model parameter $\epsilon$ to fine-tune the initial nuisance models $\mu$ and $g$ for solving the estimating equation. For a long EIC expression in Equations (7) to (9), TMLE would need to be applied to the models at each time step

separately. It has been found even in the simple static setting of the average treatment effect (ATE) estimation (with only one $\mu$ and $g$ model) [21] that the performance of TMLE is dependent on the quality of the initial models, and tuning the initial models and $\epsilon$ jointly can lead to more robust performance. To improve the quality of the initial models, one can also consider using cross-validation based methods such as super learning [50] that constructs an initial model by combining multiple machine learning models. However, the problem of TMLE is more severe in DTR settings which involves a relatively large number of initial models over time. Also, it is computationally prohibitive to use cross-validation repeatedly when we are optimizing a DTR.

## 4.3 Model

Next, we present our method, GRV, that regularizes all the nuisance models with a fluctuation parameter $\epsilon_t$ to solve the estimating equation defined by the sophisticated EIC in Equations (7) to (9) in the training time of the nuisance models. We prove theoretically that the equation is solved when the regularizer is minimized. Such a strategy simplifies the process of solving multiple estimating equations and avoids using bad initial nuisance models to construct the value function estimator.

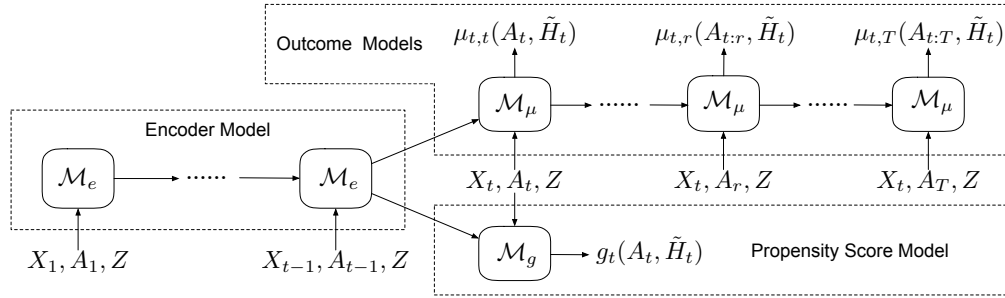

Figure 1: The network architecture of outcome and propensity score models

We start by introducing the network architecture that parameterizes the nuisance models. Our network is given as a composition of standard RNN models and feedforward neural network (i.e. multilayer perceptron) models. We opt for neural network models because they enable end-to-end training of the nuisance models. Figure 1 gives a pictorial description of the network architecture. We use a sequence encoder model $\mathcal{M}_e$ to extract information from the history $H_{t-1}$. At each time step $s$, the encoder $\mathcal{M}_e$ takes the covariates $X_s$, treatment variables $A_s$, and baseline covariates $Z$ concatenated together as its input. We feed in the baseline covariates repeatedly for learning its interaction with the covariate and treatment variables at each time step. At time $t$, we use a sequence decoder model $\mathcal{M}_\mu$ to parametrize the outcome models $\mu_{t,r}(A_{t:r}, \tilde{H}_t), r = t, \ldots, T$. The decoder $\mathcal{M}_\mu$ takes the hidden states of $\mathcal{M}_e$ at time $t-1$ as its initial state. The time $r$ input of $\mathcal{M}_\mu$ consists of $X_t$, $A_r$ and $Z$. The propensity score models $g_s(A_s, \tilde{H}_s), s = t, \ldots, T$, are parameterized by a shared multilayer perceptron. The model $g_t(A_t, \tilde{H}_t)$ takes the hidden states of $\mathcal{M}_e$ at time $t-1$, $X_t$ and $Z$ as input, and use a softmax output layer to estimate the probability of $A_t$. We note that using a shared encoder is a common strategy in the recent neural network literature for treatment effect estimation. However, such a strategy is only beneficial when the outcomes and propensity scores are some similar functions, e.g., functions with the same dependence on the input covariates. On the contrary, if the outcomes and propensity scores are some different functions, we should model them with two separate encoders. For example, the treatment assignment is randomized while the outcomes have some complex dependence on the input covariates.

**Objective function.** To simplify the notation in our objective function, we define $\hat{\delta}_{i,s:r} = \prod_{m=s}^{r} \mathbb{1}\big(A_{i,m} = \hat{d}_m(\tilde{H}_{i,m})\big)$, $g_{i,s:r} = \prod_{m=s}^{r} g_m(A_{i,m}, \tilde{H}_{i,m})$, and $\hat{d}_{i,s:r} = \big[\hat{d}_m(\tilde{H}_{i,m})\big]_{m=s}^{r}$. Given the treatment rules $\hat{d}_{t:T}$, we optimize $\mathcal{M}_{\text{NN}} = \{\mathcal{M}_e, \mathcal{M}_\mu, \mathcal{M}_g\}$ with the objective function

$$\hat{\mathcal{M}}_{\text{NN}}, \hat{\epsilon}_t = \underset{\mathcal{M}_{\text{NN}}, \epsilon_t}{\arg\min} \Big\{ \mathcal{L}_t\big(\mathcal{M}_{\text{NN}}; O_{[N]}\big) + \mathcal{R}_t\big(\mathcal{M}_{\text{NN}}, \epsilon_t; \hat{d}_{t:T}, O_{[N]}\big) \Big\}, \tag{10}$$

where $\mathcal{L}_t\big(\mathcal{M}_{\mathrm{NN}}; O_{[N]}\big)$ is given as

$$\mathcal{L}_t\big(\mathcal{M}_{\mathrm{NN}}; O_{[N]}\big) = \sum_{s=t}^{T} \left( \frac{1}{N} \sum_{i=1}^{N} \big[\mu_{s,s}(A_{i,s}, \tilde{H}_{i,s}) - Y_{i,s}\big]^2 + \frac{1}{N} \sum_{i=1}^{N} \mathrm{CrossEntropy}\big[g_{i,s}, A_{i,s}\big] \right),$$

where the first term is the loss for the outcome models, and the second term is the loss for the propensity score models. The regularizer $\mathcal{R}_t\big(\mathcal{M}_{\mathrm{NN}}, \epsilon_t; \hat{d}_{t:T}, O_{[N]}\big)$ is factorized as $\mathcal{R}_t\big(\mathcal{M}_{\mathrm{NN}}, \epsilon_t; \hat{d}_{t:T}, O_{[N]}\big) = \sum_{r=t}^{T} \sum_{s=t}^{r} \zeta_{s,r}^t$, where each factor $\zeta_{s,r}^t$ is defined as

$$\zeta_{s,r}^t = \frac{1}{N} \sum_{i=1}^{N} \big[\mu_{s+1:r}^{\epsilon_t}\big(\hat{d}_{i,s+1:r}, \tilde{H}_{i,s+1}\big) - \mu_{s:r}^{\epsilon_t}\big(\hat{d}_{i,s:r}, \tilde{H}_{i,s}\big)\big]^2, \text{ as } s = t, \ldots, r-1,$$

and $\zeta_{r,r}^t = \frac{1}{N} \sum_{i=1}^{N} \big[Y_{i,r} - \mu_{r,r}^{\epsilon_t}(\hat{d}_{i,r}, \tilde{H}_{i,r})\big]^2$, where $\mu_{k,r}^{\epsilon_t}$ is a fluctuated outcome model,

$$\mu_{k,r}^{\epsilon_t}\big(\hat{d}_{i,k:r}, \tilde{H}_{i,k}\big) = \mu_{k,r}\big(\hat{d}_{i,k:r}, \tilde{H}_{i,k}\big) + \epsilon_t \sum_{m=k}^{r} \frac{\hat{\delta}_{i,t:m}}{g_{i,t:m}}, \text{ for } r = t, \ldots, T, \ k = t, .., r.$$

The following theorem shows that minimizing $\mathcal{R}_t\big(\mathcal{M}_{\mathrm{NN}}, \epsilon_t; \hat{d}_{t:T}, O_{[N]}\big)$ in Equation (10) encourages these fluctuated outcome models to solve the estimating equation, hence the resulting estimator is tailored to be efficient for estimating the value function $V_t(\hat{d}_{t:T})$ as discussed in Section 4.2. Since we already adapt all the nuisance models to solve the estimating equation, we have the freedom to scale or penalize the magnitude of $\epsilon_t$ to be small in training, which can partially relieve the high variance issue caused by the unstable IPS product. We note that minimizing $\mathcal{L}_t\big(\mathcal{M}_{\mathrm{NN}}; O_{[N]}\big)$ is still important for the convergence of the remainder in Equation (6).

**Theorem 1.** *Given some treatment rules $\hat{d}_{t:T} = \{\hat{d}_t, \ldots, \hat{d}_T\}$, the nuisance models $\hat{\mu}$ and $\hat{g}$ parameterized by $\hat{\mathcal{M}}_{NN}$ and $\hat{\epsilon}_t$ satisfy the estimating equation when $\mathcal{R}_t$ is minimized such that*

$$0 = \partial_{\epsilon_t}\Big[\mathcal{R}_t\big(\hat{\mathcal{M}}_{NN}, \epsilon_t; \hat{d}_{t:T}, O_{[N]}\big)\Big]\Big|_{\epsilon_t = \hat{\epsilon}_t} = \mathbb{P}_N D_t^*(\hat{P}_{t:T}; \hat{d}_{t:T}). \tag{11}$$

*Proof.* Appendix A. □

Recall that $\hat{\mu}_{t,r}^{\hat{\epsilon}_t}\big(d_{t:r}, \tilde{H}_t\big)$ estimates $\mathbb{E}\big[Y_r(d_{t:r}) \mid \tilde{H}_t\big]$, the mean of $Y_r$ given the intervention of $d_{t:r}$ and the history $\tilde{H}_t$. We can construct an empirical estimate $\hat{V}_t(d_{t:T})$ of the time $t$ value function $\hat{V}_t(d_{t:T})$ by summing up the estimated outcomes over time. We can learn each $d_t$ backwardly in a dynamic programming fashion, by optimizing the estimator of $\hat{V}_t(d_t, \hat{d}_{t+1:T})$ given the previously optimized rules $\hat{d}_{t+1:T}$. We also attempt to optimize all the treatment rules in $d$ jointly. We denote the former DTR optimization algorithm as GRV-B and the latter as GRV-S. Both algorithms are computationally expensive because we need to update the nuisance models after each update of the DTR. Our hope is that training the DTR for one epoch would not change the DTR dramatically, then we only need to retrain the nuisance models for a few epochs. GRV-S is cheaper than GRV-B but unstable in practice. Finally, we note that optimizing efficient value function estimates is a popular strategy for learning treatment rules in the literature. However, even if the estimates are efficient, it does not imply that we have an efficient learning algorithm of the optimal DTR. It has been found recently that efficient gradient estimation is crucial for achieving efficient DTR learning [56, 57, 58].

## 5 Experiments

We evaluate the performance of GRV[1] in DTR evaluation and optimization. The first set of our experiments is based on two non-Markovian simulation studies adapted from [43]. We call them treatment cost trade-off and survival rate maximization respectively. In the first simulation, the DTR objective is to minimize the total treatment cost in the treatment trajectory while keeping the health metric above the threshold at the end of the trajectory. In the second simulation, the DTR objective is to maximize the survival rate of cancer patients using an invasive and non-invasive treatment in

Table 1: Performance of benchmarks, our estimator and algorithms: the MSEs of the value function estimators (lower is better) and the value of the learned DTRs (higher is better). The mean and standard deviation are averaged across 20 independent runs on a testing set with 20,000 individuals.

| *Metrics* | MSE | | | | Value | | | |
|---|---|---|---|---|---|---|---|---|
| *Dataset* | Treatment cost trade-off | | | | | | | |
| *Samples* | IPW | BQL | AIPW | GRV | IPW | BQL | GRV-S | GRV-B |
| 1000 | $.375 \pm .018$ | $.362 \pm .003$ | $.139 \pm .006$ | $\mathbf{.134 \pm .004}$ | $.741 \pm .002$ | $.748 \pm .002$ | $.753 \pm .004$ | $\mathbf{.761 \pm .003}$ |
| 5000 | $.223 \pm .011$ | $.348 \pm .003$ | $.035 \pm .004$ | $\mathbf{.031 \pm .003}$ | $.760 \pm .002$ | $.752 \pm .001$ | $.759 \pm .003$ | $\mathbf{.767 \pm .002}$ |
| 10000 | $.132 \pm .013$ | $.291 \pm .002$ | $\mathbf{.023 \pm .003}$ | $\mathbf{.022 \pm .002}$ | $.765 \pm .001$ | $.754 \pm .001$ | $.762 \pm .002$ | $\mathbf{.768 \pm .002}$ |
| *Dataset* | Survival rate maximization | | | | | | | |
| *Samples* | IPW | BQL | AIPW | GRV | IPW | BQL | GRV-S | GRV-B |
| 1000 | $.332 \pm .035$ | $.034 \pm .001$ | $.047 \pm .005$ | $\mathbf{.032 \pm .002}$ | $.154 \pm .009$ | $.137 \pm .006$ | $.146 \pm .005$ | $\mathbf{.182 \pm .004}$ |
| 5000 | $.065 \pm .003$ | $.017 \pm .001$ | $.006 \pm .002$ | $\mathbf{.005 \pm .001}$ | $.175 \pm .008$ | $.198 \pm .003$ | $.201 \pm .002$ | $\mathbf{.224 \pm .002}$ |
| 10000 | $.031 \pm .004$ | $.011 \pm .001$ | $\mathbf{.003 \pm .001}$ | $\mathbf{.003 \pm .001}$ | $.188 \pm .006$ | $.215 \pm .002$ | $.213 \pm .002$ | $\mathbf{.226 \pm .002}$ |

the treatment trajectories. We benchmark GRV against three methods that cover the three estimator classes summarized in Section 3, including IPW/IS estimator [26, 27], batch $Q$-learning (BQL) [28, 33], and AIPW estimator [40, 42]. The simulation and implementation details of GRV and benchmark methods can be found in Appendix B.

Given some offline data from a simulation environment and a target DTR, we want to test how accurate the methods can estimate the value of the target DTR. We compare the methods in terms of their mean squared errors (MSEs) in estimating the value of the optimal DTR. For DTR learning, we compare GRV-S and GRV-B against the benchmarks in terms of the value achieved by their learned DTRs. The DTR value is defined as the cumulative outcome under the learned DTR. Specifically, we generate a large testing dataset with 20,000 individuals. For each learned DTR, we let each individual in the testing set follow the treatments recommended by the DTR. At the end of the treatment trajectory, we obtain the cumulative outcome for each individual. We then compute the value of the DTR by averaging the cumulative outcome over all the individuals.

In Table 1, we report the MSEs of the value function estimators and the value of the learned DTRs. We repeat the same experiments with training sample size at 1000, 5000 and 10000. When the sample size is 1000 or 5000, our GRV estimator outperforms the competing estimators in estimating the value of the optimal DTR. When the sample size is 10000, AIPW and GRV achieve similar MSE performance. Additionally, the DTRs learned by GRV-B achieve larger value than the DTRs learned by the other methods. GRV-S performs relatively poorly because optimizing all the treatment rules together is unstable when the trajectory is long. Overall, the performance gain highlights the effectiveness of our regularizer, which constructs an efficient value function estimator by adapting the nuisance models to solve the estimating equation during the training process of the nuisance models.

In Appendix C, we show our method performs well on three additional simulation studies from [6] where the covariate vector is relatively high-dimensional while the training set is small with hundreds of samples. We also provide an ablation study which shows that ablating the regularizer leads to worse performance in our method. In Appendix D, we demonstrate the application of our method to antibiotics administration on a real-world dataset extracted from the MIMIC database [22]. We note that this experiment is only used to illustrate that the learned DTR will lead to gradually decrease the usage of the antibiotics treatment, which corresponds to the clinical literature. The real DTR application can only be demonstrated well through real-world experiments and simulation. For example, in a real-world test-bed, the clinician can check if his/her treatment decision is correct or not by comparing it with the DTR recommendation before prescribing it.

## 6 Conclusion

We have introduced Gradient Regularized $V$-Learning (GRV), a novel regularization method that enables recurrent neural network models to estimate the value function of a target DTR accurately and learn better DTRs. We prove theoretically that the nuisance models satisfy the estimating equation in semiparametric estimation theory when the proposed regularizer is minimized. We hope that GRV will become a useful regularization method when RNNs are deployed to tackle the challenges of treatment effect estimation and decision making in machine learning.

## Broader Impact

Our work can help to develop accurate and individualized decision-making system in many real-world applications, such as treatment recommendation. Our method is easy to use for machine learning practitioners since it simply relies on regularization of the underlying recurrent network models. However, Offline DTR or policy evaluation is still a challenging problem because we often do not have sufficient samples to estimate the outcomes for all the possible treatment plans over time. Given some datasets that are high-dimensional or have long treatment trajectories, we would need to combine methods that solve different challenges of DTR evaluation, such as putting constraints on treatment assignment mechanisms over time or selecting variables that are most likely to affect the treatment decisions. Our work focuses on a particular aspect of DTR evaluation and does not cover other aspects that are also important for the real-world applications of DTRs.

## Acknowledgments

This work was supported by GlaxoSmithKline (GSK), the US Office of Naval Research (ONR), and the National Science Foundation (NSF) 1722516. We thank all reviewers for their generous comments and suggestions.

## Footnotes

[1]The code is provided at: https://bitbucket.org/mvdschaar/mlforhealthlabpub.

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
