[Supplementary Material · grv_appendix.pdf]

# Appendix: Gradient Regularized $V$-Learning for Dynamic Treatment Regimes

## A  Proof of Theorem 1

**Theorem 1.** *Given some treatment rules $\hat{d}_{t:T} = \{\hat{d}_t, \ldots, \hat{d}_T\}$, the nuisance models $\hat{\mu}$ and $\hat{g}$ parameterized by $\hat{\mathcal{M}}_{NN}$ and $\hat{\epsilon}_t$ satisfy the estimating equation when $\mathcal{R}_t$ is minimized such that*

$$0 = \partial_{\epsilon_t}\Big[\mathcal{R}_t\big(\hat{\mathcal{M}}_{NN}, \epsilon_t; \hat{d}_{t:T}, O_{[N]}\big)\Big]\Big|_{\epsilon_t = \hat{\epsilon}_t} = \mathbb{P}_N D_t^*(\hat{P}_{t:T}; \hat{d}_{t:T}). \tag{12}$$

*Proof.* With the treatment rules $\hat{d}_{t:T}$, we defne a set of fluctuated outcome models,

$$
\begin{aligned}
\hat{\mu}_{k,r}^{\epsilon_t}\big(\hat{d}_{k:r}, \tilde{H}_k\big) &= \hat{\mu}_{k,r}\big(\hat{d}_{k:r}, \tilde{H}_k\big) + \epsilon_t \sum_{m=k}^{r} \frac{\hat{\delta}_{t:m}}{\hat{g}_{t:m}} \\
&= \hat{\mu}_{k,r}\big(\hat{d}_{k:r}, \tilde{H}_k\big) + \epsilon_t \sum_{m=k}^{r} \frac{\prod_{l=t}^{m} \mathbb{1}(A_l = d_l(\tilde{H}_l))}{\prod_{l=t}^{m} \hat{g}_l(A_l, \tilde{H}_l)},
\end{aligned}
\tag{13}
$$

for $r = t, \ldots, T$ and $k = t, .., r$. The regularizer $\mathcal{R}\big(\mathcal{M}_{NN}, \epsilon; \hat{d}_{t:T}, O_{[N]}\big)$ is given as

$$\mathcal{R}\big(\mathcal{M}_{NN}, \epsilon_t; \hat{d}_{t:T}, O_{[N]}\big) = \sum_{r=t}^{T} \sum_{s=t}^{r} \zeta_{s,r}^t.$$

When $s = t, \ldots, r-1$,

$$
\begin{aligned}
\zeta_{s,r}^t &= \frac{1}{N} \sum_{i=1}^{N} \Big[\hat{\mu}_{s+1:r}^{\epsilon_t}\big(\hat{d}_{i,s+1:r}, \tilde{H}_{i,s+1}\big) - \hat{\mu}_{s:r}^{\epsilon_t}\big(\hat{d}_{i,s:r}, \tilde{H}_{i,s}\big)\Big]^2 \\
&= \frac{1}{N} \sum_{i=1}^{N} \Big[\Big(\hat{\mu}_{s+1,r}\big(\hat{d}_{i,s+1:r}, \tilde{H}_{i,s+1}\big) + \epsilon_t \sum_{m=s+1}^{r} \frac{\hat{\delta}_{i,t:m}}{\hat{g}_{i,t:m}}\Big) - \Big(\hat{\mu}_{s,r}\big(\hat{d}_{i,s:r}, \tilde{H}_{i,s}\big) + \epsilon_t \sum_{m=s}^{r} \frac{\hat{\delta}_{i,t:m}}{\hat{g}_{i,t:m}}\Big)\Big]^2 \\
&= \frac{1}{N} \sum_{i=1}^{N} \Big[\hat{\mu}_{s+1,r}\big(\hat{d}_{i,s+1:r}, \tilde{H}_{i,s+1}\big) - \hat{\mu}_{s,r}\big(\hat{d}_{i,s:r}, \tilde{H}_{i,s}\big) - \epsilon_t \frac{\hat{\delta}_{i,t:s}}{\hat{g}_{i,t:s}}\Big]^2,
\end{aligned}
$$

and as $s = r$,

$$
\begin{aligned}
\zeta_{r,r}^t &= \frac{1}{N} \sum_{i=1}^{N} \Big[Y_{i,r} - \hat{\mu}_{r,r}^{\epsilon_t}\big(\hat{d}_{i,r}, \tilde{H}_{i,r}\big)\Big]^2 \\
&= \frac{1}{N} \sum_{i=1}^{N} \Big[Y_{i,r} - \hat{\mu}_{r,r}\big(\hat{d}_{i,r}, \tilde{H}_{i,r}\big) - \epsilon_t \frac{\hat{\delta}_{i,t:r}}{\hat{g}_{i,t:r}}\Big]^2.
\end{aligned}
$$

Let $\hat{\epsilon}_t$ denote the minimizer of $\mathcal{R}_t\big(\hat{\mathcal{M}}_{NN}, \epsilon_t; \hat{d}_{t:T}, O_{[N]}\big)$ where the partial derivative of $\mathcal{R}_t\big(\hat{\mathcal{M}}_{NN}, \epsilon_t; \hat{d}_{t:T}, O_{[N]}\big)$ w.r.t $\epsilon_t$ equals to zero. When $\epsilon_t = \hat{\epsilon}_t$,

$$
\begin{aligned}
0 &= \partial_{\epsilon_t}\Big[\mathcal{R}\big(\hat{\mathcal{M}}_{NN}, \epsilon_t; \hat{d}_{t:T}, O_{[N]}\big)\Big]\Big|_{\epsilon_t = \hat{\epsilon}_t} \\
&= \sum_{r=t}^{T} \sum_{s=t}^{r} \partial_{\epsilon_t}\big[\zeta_{s,r}^t\big]\big|_{\epsilon_t = \hat{\epsilon}_t} \\
&= \sum_{r=t}^{T} \Bigg(\sum_{s=t}^{r-1} \frac{1}{N} \sum_{i=1}^{N} \frac{\hat{\delta}_{i,t:s}}{\hat{g}_{i,t:s}}\Big[\hat{\mu}_{s+1,r}\big(\hat{d}_{i,s+1:r}, \tilde{H}_{i,s+1}\big) - \hat{\mu}_{s,r}\big(\hat{d}_{i,s:r}, \tilde{H}_{i,s}\big) - \hat{\epsilon}_t \frac{\hat{\delta}_{i,t:s}}{\hat{g}_{i,t:s}}\Big] \\
&\quad + \frac{1}{N} \sum_{i=1}^{N} \frac{\hat{\delta}_{i,t:r}}{\hat{g}_{i,t:r}}\Big[Y_{i,r} - \hat{\mu}_{r,r}\big(\hat{d}_{i,r}, \tilde{H}_{i,r}\big) - \hat{\epsilon}_t \frac{\hat{\delta}_{i,t:r}}{\hat{g}_{i,t:r}}\Big]\Bigg).
\end{aligned}
\tag{14}
$$

The sample-average EIC, $\mathbb{P}_N D_t^*(\hat{P}_{t:T}; \hat{d}_{t:T})$, takes the same form as Equations (7) to (9),

$$\mathbb{P}_N D_t^*(\hat{P}_{t:T}; \hat{d}_{t:T}) = \sum_{r=t}^{T} \frac{1}{N} \sum_{i=1}^{N} \left( \mu_{t,r}^{\hat{\epsilon}_t}(\hat{d}_{i,t:r}, \tilde{H}_{i,t}) - \frac{1}{N} \sum_{j=1}^{N} \mu_{t,r}^{\hat{\epsilon}_t}(\hat{d}_{j,t:r}, \tilde{H}_{j,t}) \right)$$

$$+ \sum_{r=t}^{T} \sum_{s=t}^{r} \mathbb{P}_N D_{s,r}^*(\hat{P}_{t:T}; \hat{d}_{t:T}). \tag{15}$$

We first note that the first term in Equation (15) vanishes as

$$\sum_{r=t}^{T} \frac{1}{N} \sum_{i=1}^{N} \left( \mu_{t,r}^{\hat{\epsilon}_t}(\hat{d}_{i,t:r}, \tilde{H}_{i,t}) - \frac{1}{N} \sum_{j=1}^{N} \mu_{t,r}^{\hat{\epsilon}_t}(\hat{d}_{j,t:r}, \tilde{H}_{j,t}) \right)$$

$$= \sum_{r=t}^{T} \frac{1}{N} \sum_{i=1}^{N} \mu_{t,r}^{\hat{\epsilon}_t}(\hat{d}_{i,t:r}, \tilde{H}_{i,t}) - \sum_{r=t}^{T} \frac{1}{N} \sum_{j=1}^{N} \mu_{t,r}^{\hat{\epsilon}_t}(\hat{d}_{j,t:r}, \tilde{H}_{j,t}) \tag{16}$$

$$= 0.$$

When $s = t, \ldots, r-1$, we know from Equation (8) that

$$\mathbb{P}_N D_{s,r}^*(\hat{P}_{t:T}; \hat{d}_{t:T})$$

$$= \frac{1}{N} \sum_{i=1}^{N} \frac{\hat{\delta}_{i,t:s}}{\hat{g}_{i,t:s}} \left[ \hat{\mu}_{s+1,r}^{\hat{\epsilon}_t}(\hat{d}_{i,s+1:r}, \tilde{H}_{i,s+1}) - \hat{\mu}_{s,r}^{\hat{\epsilon}_t}(\hat{d}_{i,s:r}, \tilde{H}_{i,s}) \right]$$

$$= \frac{1}{N} \sum_{i=1}^{N} \frac{\hat{\delta}_{i,t:s}}{\hat{g}_{i,t:s}} \left[ \left( \hat{\mu}_{s+1,r}(\hat{d}_{i,s+1:r}, \tilde{H}_{i,s+1}) + \hat{\epsilon}_t \sum_{m=s+1}^{r} \frac{\hat{\delta}_{i,t:m}}{\hat{g}_{i,t:m}} \right) - \left( \hat{\mu}_{s,r}(\hat{d}_{i,s:r}, \tilde{H}_{i,s}) + \hat{\epsilon}_t \sum_{m=s}^{r} \frac{\hat{\delta}_{i,t:m}}{\hat{g}_{i,t:m}} \right) \right]$$

$$= \frac{1}{N} \sum_{i=1}^{N} \frac{\hat{\delta}_{i,t:s}}{\hat{g}_{i,t:s}} \left[ \hat{\mu}_{s+1,r}(\hat{d}_{i,s+1:r}, \tilde{H}_{i,s+1}) - \hat{\mu}_{s,r}(\hat{d}_{i,s:r}, \tilde{H}_{i,s}) - \hat{\epsilon}_t \frac{\hat{\delta}_{i,t:s}}{\hat{g}_{i,t:s}} \right].$$

When $s = r$, we know from Equation (9) that

$$\mathbb{P}_N D_{r,r}^*(\hat{P}_{t:T}; \hat{d}_{t:T}) = \frac{1}{N} \sum_{i=1}^{N} \frac{\hat{\delta}_{i,t:r}}{\hat{g}_{i,t:r}} \left( Y_{i,r} - \hat{\mu}_{r,r}^{\hat{\epsilon}_t}(\hat{d}_{i,r}, \tilde{H}_{i,r}) \right)$$

$$= \frac{1}{N} \sum_{i=1}^{N} \frac{\hat{\delta}_{i,t:r}}{\hat{g}_{i,t:r}} \left( Y_{i,r} - \hat{\mu}_{r,r}(\hat{d}_{i,r}, \tilde{H}_{i,r}) - \hat{\epsilon}_t \frac{\hat{\delta}_{i,t:r}}{\hat{g}_{i,t:r}} \right).$$

With the expressions above, we can rewrite Equation (15) explicitly as

$$\mathbb{P}_N D_t^*(\hat{P}_{t:T}; \hat{d}_{t:T}) = 0 + \sum_{r=t}^{T} \left( \sum_{s=t}^{r-1} \frac{1}{N} \sum_{i=1}^{N} \frac{\hat{\delta}_{i,t:s}}{\hat{g}_{i,t:s}} \left[ \hat{\mu}_{s+1,r}(\hat{d}_{i,s+1:r}, \tilde{H}_{i,s+1}) - \hat{\mu}_{s,r}(\hat{d}_{i,s:r}, \tilde{H}_{i,s}) \right. \right.$$

$$\left. \left. - \hat{\epsilon}_t \frac{\hat{\delta}_{i,t:s}}{\hat{g}_{i,t:s}} \right] + \frac{1}{N} \sum_{i=1}^{N} \frac{\hat{\delta}_{i,t:r}}{\hat{g}_{i,t:r}} \left[ Y_{i,r} - \hat{\mu}_{r,r}(\hat{d}_{i,r}, \tilde{H}_{i,r}) - \hat{\epsilon}_t \frac{\hat{\delta}_{i,t:r}}{\hat{g}_{i,t:r}} \right] \right),$$

which equals to the R.H.S of Equation (14), which establishes the equality in Equation (12). Finally, we note that even if we scale down $\epsilon_t$ by dividing it with a large constant in the fluctuated models in Equation (13), the same proof still holds by multiplying Equation (14) by the large constant. □

## B  Experiment details

In this section, we first introduce the model architecture and pseudo code of GRV-S and GRV-B in DTR learning. Then we will provide the implementation and simulation details of our experiments. We note that the model architectures and pseudo codes introduced later are generic and used for the additional experiments in Appendix C. In the simulations of the main manuscript, we focus on developing when-to-treat or when-to-stop DTR. In these simulations, we can only train the time $t$ treatment rule on the samples that are not treated before time $t$, because these simulations only allow one treatment action to be made throughout a trajectory. The multi-step treatment decision problem reduces to decide when to treat an individual or when to stop the treatment for an individual.

## B.1 GRV-S

**Model architecture.** In GRV-S, we optimize the treatment rules in a DTR $d$ jointly. We use a shared neural network model to parameterize the treatment rules $d_t$, $t \in [T]$, as shown in Figure 2. The model $\mathcal{M}_d$ is a sequence model, e.g. a vanilla RNN, concatenated with a shared multilayer perceptron (MLP). The sequence model $\mathcal{M}_d$ has the same input as the encoder $\mathcal{M}_e$ in Figure 1 of the main manuscript. The inputs $X_0$ and $A_0$ are zero vectors. At time $t$, the MLP takes the RNN hidden states at time $t-1$ and $X_t$ as input, and use a softmax output layer to generate a $K$-dimensional probability vector. The treatment option $d_t(\tilde{H}_t)$ is a one-hot vector randomly sampled w.r.t the probability vector.

Figure 2: The DTR network in GRV-S

---

**Algorithm 1** GRV-S

---

**Input:** A observational dataset $O_{[N]} = \left\{ O_i = (Z_i, X_{i,[T]}, A_{i,[T]}, Y_{i,[T]}), i \in [N] \right\}$, and the maximum number of iterations, $N_{\text{opt}}$
**Initialization:** Randomly initialize the DTR network in Figure 2, the value function network $\mathcal{M}_{\text{NN}}$ in Figure 1, and the fluctuation parameter $\epsilon_1$
**for** $n = 1$ **to** $N_{\text{opt}}$ **do**
    Sample the DTR decisions $\hat{d}_{i,1:T}$, for each unit $i \in [N]$
    Optimize $\epsilon_1$ and the value function network $\mathcal{M}_{\text{NN}}$ based on Equation (10) with $t = 1$
    Construct the empirical value function $\hat{V}(d)$ with $(\hat{\mathcal{M}}_{\text{NN}}, \hat{\epsilon}_1)$
    Update the entire DTR $\hat{d}$ by optimizing the empirical estimate $\hat{V}(d)$
**end for**
**Output:** DTR $\hat{d} = \{\hat{d}_1, \ldots, \hat{d}_T\}$

---

## B.2 GRV-B

**Model architecture.** We optimize the treatment rules backwardly through time. The optimized rule $\hat{d}_t$ will be used in the optimization of the rules in the earlier time steps, $d_s, s = t-1, \ldots, 1$. We let an independent neural network model to parameterize each treatment rule. The treatment rule $d_t$ is parameterized by the network in Figure 3. The architecture is almost the same as the one used in GRV-S, but we only have one output at time $t$. The network is given as a sequence model $\mathcal{M}_{d_t}$, e.g. a vanilla RNN, concatenated with a multilayer perceptron (MLP). At time $t$, the MLP takes the RNN hidden states at time $t-1$ and $X_t$ as input and uses a softmax output layer to generate a $K$-dimensional probability vector. The treatment option $d_t(\tilde{H}_t)$ is a one-hot vector randomly sampled w.r.t the probability vector. If the dataset is low-dimensional, we can also parameterize each treatment rule using a standard supervised learning model which takes all the time-varying covariates, treatment variables and baseline covariates in the history as input.

Figure 3: The network of the treatment rule $d_t$ in GRV-B

---

**Algorithm 2** GRV-B

---

**Input:** A observational dataset $O_{[N]} = \{O_i = (Z_i, X_{i,[T]}, A_{i,[T]}, Y_{i,[T]}), i \in [N]\}$, and the maximum number of iterations, $N_{\text{opt}}$
**Initialization:** Randomly initialize the DTR network in Figure 3, the value function network $\mathcal{M}_{\text{NN}}$ in Figure 1, and the fluctuation parameters $\epsilon_t, t \in [T]$
**for** $n = 1$ **to** $N_{\text{opt}}$ **do**
   Sample the DTR decisions $\hat{d}_{i,T}$, for each unit $i \in [N]$
   Optimize $\epsilon_T$ and the value function network $\mathcal{M}_{\text{NN}}$ based on Equation (10) with time step $T$
   Construct the empirical value function $\hat{V}_T(d_T)$ with $(\hat{\mathcal{M}}_{\text{NN}}, \hat{\epsilon}_T)$
   Update the treatment rule $\hat{d}_T$ by optimizing the empirical estimate $\hat{V}_T(d_T)$
**end for**
**for** $t = T - 1$ **to** $1$ **do**
   Reinitialized the value function network $\mathcal{M}_{\text{NN}}$
   **for** $n = 1$ **to** $N_{\text{opt}}$ **do**
      Sample the DTR decisions $\hat{d}_{i,t:T}$, for each unit $i \in [N]$
      Optimize $\epsilon_t$ and the value function network $\mathcal{M}_{\text{NN}}$ based on Equation (10) with time step $t$
      Construct the empirical value function $\hat{V}_{t:T}(d_t, \hat{d}_{t+1:T})$ with $(\hat{\mathcal{M}}_{\text{NN}}, \hat{\epsilon}_t)$
      Update the treatment rule $\hat{d}_t$ by optimizing the empirical estimate $\hat{V}_{t:T}(d_t, \hat{d}_{t+1:T})$
   **end for**
**end for**
**Output:** DTR $\hat{d} = \{\hat{d}_1, \ldots, \hat{d}_T\}$

---

### B.3 Implementation

We use GRU cells for all our RNN models. The MLP models concatenated with the RNNs and the propensity score network are two-layer SELU networks. Because our dataset is not high-dimensional, we set the number of units in the hidden layers to 32 for all the models and regularize the RNN models by setting the dropout rate to 0.5. We warm start the training by minimizing the first term $\mathcal{L}(\mathcal{M}_{\text{NN}}; O_{[N]})$ in Equation (10) which does not depend on the DTR. In training, the learning rate is 0.001, and the maximum number of iterations $N_{\text{opt}}$ is 100. At each iteration, we train the treatment rule in GRV-B or the entire DTR in GRV-S for one epoch, then we retrain the network $\mathcal{M}_{\text{NN}}$ and the fluctuation parameter $\epsilon_t$ for five epochs. One epoch means one loop over all the batch samples in the training set. We implement the benchmarks following the code provided in the R package[2] of [43].

### B.4 Simulation

We introduce the data generating processes in the two simulation studies [43] that we use in the experiment section. For each study, we show how each variable is generated in the environments, how the treatment and covariates influence the outcomes directly or indirectly, which variables we observe, and why the environment is non-Markovian.

**Treatment Cost Trade-off.** We consider a setting where we track a health metric and get a reward if the health metric is above a threshold at $T = 10$. The treatment provides a positive nudge to the health metric at a cost. We start with treatment on, and need to choose when to stop to minimize cost

while trying to keep the health metric stay above the threshold. The data is generated as follows,

$$X_1 \sim N(0,1), \ X_{t+1} \mid X_t, A_t \sim \mathbb{1}_{X_t \geq -0.5} N \left( X_t + \frac{1 - A_t}{1 + \exp(0.3 X_t)}, \frac{1}{2T} \right) + \mathbb{1}_{X_t < -0.5} X_t,$$

$$\tilde{X}_t \sim N(X_t, 0.25),$$

$$Y_t = -\frac{1}{T}(1 - A_t), \text{for } t = 1, .., T - 1,$$

$$Y_T = \mathbb{1}_{\tilde{X}_{T+1} > 0} - \frac{1}{T}(1 - A_t),$$

with the stopping action,

$$A_t \mid X_t \sim \text{Bern} \left( 1 / \left[ 1 + \exp(1.5 - X_t) + \exp(3 - t) \right] \right).$$

In this simulation, we want to stop the treatment optimally to maximize the treatment cost trade-off,

$$\sum_{t=1}^{T} Y_t = \mathbb{1}_{\tilde{X}_{T+1} > 0} - \frac{1}{T} \sum_{t=1}^{T} A_t.$$

From the equations above, we can see if the treatment is on ($A_t = 0$), we would get a cost $-1/T$ on the outcome $Y_t$. However, in the generating process of $X_{t+1}$ from $X_t$ and $A_t$, $A_t = 0$ can help to increase the value of $X_{t+1}$. However, this increase is small when $X_t$ is large. If $\tilde{X}_{T+1}$ is larger than zero, we would bet a reward in $Y_T$. Roughly speaking, the optimal strategy is to stop the treatment at some points because the treatment cost is the same while the treatment influence on $X_{t+1}$ and $\tilde{X}_{T+1}$ decreases over time. Finally, we note that we do not assume Markovian structure in the simulation and only get to observe a noisy version $\tilde{X}_t$ of the true covariate vector $X_t$ at time $t$. The action generation is also time-dependent and non-stationary over time.

**Survival Rate Maximization.** In the second setup, we consider multiple treatment choices. Our design here is motivated by a healthcare setting where once a doctor starts treatment, they can choose between a more effective but more invasive treatment with strong side effects, or a less effective but less invasive treatment. More specifically, imagine a cancer patient's state at time $t$ is modelled by $X_{1,t}, X_{2,t}$ and $Z$ where $X_{1,t}$ is the general health state, $X_{2,t}$ is the state of a tumour, and $Z$ is not time-dependent but models the category of the patients for which lifespan differs. In particular, if $Z = 0$, a patient always dies immediately; if $Z = 1$, a patient always survives until the end of a trial; if $Z = 2$, the patient's lifespan has a strong dependency on $X_{2,t}$ which we detail below. There are two treatment choices, one non-invasive ($A_t = 1$) and one invasive ($A_t = 2$). In the main manuscript, we say $A_t$ is a one-hot vector. Here, we redefine it as the corresponding categorical variable. The non-invasive option lessens the severity of the tumour, and the invasive option completely removes the tumour but exacerbates a patient's general health conditions. At every time step $t$, we receive a binary survival outcome $Y_t$ of each patient. The DTR objective is to maximize the patients' lifetime $\sum_{t=1}^{T} Y_t$. We consider horizon $T = 10$. The data generating process is given as follows,

$$X_{1,1} \sim \exp(1), \ X_{2,1} \sim 0.5 \exp(3), \ Y_1 = 1,$$

$$Z \sim \text{Multinomial}(0.3, 0.3, 0.4),$$

$$X_{1,t+1} = \begin{cases} |X_{1,t} + u_t| & , \text{if } A_t \in \{0,1\} \\ |\max(X_{1,t}^2, 1.5 X_{1,t}) + u_t - X_{1,t}| + X_{1,t} & , \text{if } A_t = 2 \end{cases}$$

$$X_{2,t+1} = \begin{cases} |X_{2,t} + 0.5 X_{1,t} + u_t| & , \text{if } A_t = 0 \\ |0.5 X_{2,t} + u_t| & , \text{if } A_t = 1 \\ 0 & , \text{if } A_t = 2 \end{cases}$$

where $u_t \sim N(0, 0.25)$. The survival outcomes over time are generated as,

$$Y_{t+1} = \begin{cases} 0 & , \text{if } Z = 1 \\ 1 & , \text{if } Z = 2 \\ \text{Bern} \left( \mathbb{1}_{X_{2,t} \leq 5} \exp(-0.02 X_{2,t}) + \mathbb{1}_{5 < X_{2,t} \leq 14} \exp(-0.06 X_{2,t}) \right) & , \text{if } Z = 3 \end{cases}$$

From the equation of $X_{1,t+1}$, we can see $A_t = 0$ (no treatment) and $A_t = 1$ (non-invasive treatment) gives the same generating process of $X_{1,t+1}$. On the contrary, when $A_t = 2$ (invasive treatment),

$X_{1,t+1}$ will increase from $X_{1,t}$, which means the patient's general health conditions get worse. From the equation of $Y_{t+1}$ above, we can see increasing $X_{2,t}$ tends to decrease the probability of survival for the patient. If the tumour state $X_{2,t}$ is larger than 14, the probability of survival is zero. The invasive treatment ($A_t = 2$) sets $X_{2,t+1}$ to zero. Then the patient will definitely survive at time $t + 2$. The non-invase treatment ($A_t = 1$) can decrease $X_{2,t+1}$ to roughly a half of $X_{2,t}$, which also increases the chance of survival for the patient.

We only get to observe the covariates corrupted by noise at each time step,

$$\tilde{X}_{1,t} = \max(0, \min(X_{1,\max}, X_1 + \nu)),$$

$$\tilde{X}_{2,t} = \max(0, \min(X_{2,\max}, X_2 + \nu)).$$

where $\nu \sim N(0,1)$, $X_{1,\max} = 10$ and $X_{2,\max} = 16$. In this setting, the treatment assignment mechanism is based on sequential randomization in the data such that there are roughly equal number of trajectories that start treating at each time with either treatment option. This second study helps to capture settings motivated by clinical trials for longitudinal studies.

## C   Additional Experiments

### C.1   Other simulation studies

As suggested by one of the reviewers, one of our DTR learning algorithms follows the same procedure as the DTR learning algorithm backward outcome weighted learning (BOWL) [6]. In this section, we replicate the three simulation studies termed as Scenario 1,2 and 3 in [6]. In all the scenarios, the treatment variable is binary and randomized with equal probability at each time step. Despite the trajectories in these scenarios are relatively short, the scenarios do capture the challenges in DTR learning, such as the 50-dimensional covariate vector in Scenario 1 and 2, and the treatment assigned in the history can influence the future outcome in all the scenarios. We refer to the original paper for the simulation details. We compare GRV-S and GRV-B against four benchmark algorithms, including $Q$-learning (QL), $L_2$ regularized $Q$-learning (RQL), $A$-learning (AL), backward outcome weighted learning (BOWL). In Table 2, we report the mean and standard deviation of the value of the learned DTR over 500 runs in each scenario. We repeat the same experiment with 100, 200 and 400 training samples. The same sample sizes are used in the original paper.

Table 2: Performance of benchmark algorithms GRV-S and GRV-B in Scenario 1,2 and 3. The mean and standard deviation are computed average 500 runs.

| Algorithms | QL | RQL | AL | BOWL | GRV-S | GRV-B |
|---|---|---|---|---|---|---|
| Samples | | | Scenario 1 | | | |
| 100 | $0.583 \pm 1.476$ | $1.928 \pm 1.533$ | $-0.650 \pm 0.991$ | $3.849 \pm 0.918$ | $6.619 \pm 0.135$ | $\mathbf{6.680 \pm 0.186}$ |
| 200 | $0.692 \pm 0.972$ | $2.831 \pm 0.972$ | $-0.298 \pm 0.984$ | $4.502 \pm 0.768$ | $6.684 \pm 0.112$ | $\mathbf{6.714 \pm 0.155}$ |
| 400 | $3.766 \pm 0.896$ | $3.859 \pm 0.897$ | $1.973 \pm 1.072$ | $5.811 \pm 0.331$ | $6.721 \pm 0.129$ | $\mathbf{6.741 \pm 0.120}$ |
| Algorithms | QL | RQL | AL | BOWL | GRV-S | GRV-B |
| Samples | | | Scenario 2 | | | |
| 100 | $1.122 \pm 0.679$ | $2.650 \pm 0.547$ | $0.369 \pm 0.318$ | $2.709 \pm 0.340$ | $2.415 \pm 0.494$ | $\mathbf{2.852 \pm 0.097}$ |
| 200 | $1.462 \pm 0.361$ | $2.857 \pm 0.248$ | $0.631 \pm 0.322$ | $2.847 \pm 0.269$ | $3.191 \pm 0.139$ | $\mathbf{3.269 \pm 0.083}$ |
| 400 | $3.395 \pm 0.042$ | $3.418 \pm 0.056$ | $2.549 \pm 0.394$ | $3.105 \pm 0.131$ | $3.454 \pm 0.057$ | $\mathbf{3.499 \pm 0.056}$ |
| Algorithms | QL | RQL | AL | BOWL | GRV-S | GRV-B |
| Samples | | | Scenario 3 | | | |
| 100 | $7.633 \pm 2.953$ | $7.765 \pm 2.669$ | $2.184 \pm 6.377$ | $\mathbf{10.231 \pm 2.563}$ | $5.899 \pm 3.318$ | $10.082 \pm 3.215$ |
| 200 | $10.762 \pm 1.846$ | $10.860 \pm 1.676$ | $7.454 \pm 4.083$ | $\mathbf{13.139 \pm 1.952}$ | $6.945 \pm 3.727$ | $12.793 \pm 2.413$ |
| 400 | $12.105 \pm 1.605$ | $12.204 \pm 1.419$ | $10.495 \pm 1.882$ | $\mathbf{14.617 \pm 1.299}$ | $9.469 \pm 3.432$ | $14.467 \pm 1.785$ |

The GRV based algorithms outperform the benchmarks significantly in Scenario 1 and 2. However, in Scenario 3, BOWL achieves better performance than GRV-S and GRV-B. In this scenario, GRV-S performs poorly, which indicates that learning all the treatment rules jointly could be difficult when the sample size is small. The failure mode of GRV-S only appears in Scenario 3 where the only nonzero outcome is observed at the end of the trajectory. This unique characteristic may increase the difficulty of DTR optimization in GRV-S. GRV-B does not have this problem because it optimizes the treatment rules sequentially. GRV-S and GRV-B perform similarly in Scenario 1 and 2, and GRV-B

performs significantly better than GRV-S in Scenario 3. When the outcomes in the future time steps have a strong dependence on the treatments in the history, we should consider using GRV-B even though it is computationally more expensive than GRV-S because the optimization of GRV-S may be unstable when the treatment rules can influence on the outcomes at multiple time steps.

## C.2 Ablation study

In the previous experiments, the loss term and the regularizer are weighted equally in the objective function in Equation (10). Here, we provide an ablation study based on Scenario 2 above. In Table 3, we compare the performance of our method with and without the regularizer. We found that the performances of GRV-S and GRV-B drop if we remove the regularizer from the objective function, which shows that the regularizer is an important part of our method.

Table 3: Ablation study of the GRV regularizer in Scenario 2.

| Algorithms | GRV-S | | GRV-B | |
|---|---|---|---|---|
| Samples | w/o | w | w/o | w |
| 100 | $2.207 \pm 0.652$ | $\mathbf{2.415 \pm 0.494}$ | $2.732 \pm 0.117$ | $\mathbf{2.852 \pm 0.097}$ |
| 200 | $3.026 \pm 0.217$ | $\mathbf{3.191 \pm 0.139}$ | $3.155 \pm 0.103$ | $\mathbf{3.269 \pm 0.083}$ |
| 400 | $3.366 \pm 0.094$ | $\mathbf{3.454 \pm 0.057}$ | $3.362 \pm 0.058$ | $\mathbf{3.499 \pm 0.056}$ |

# D  Experiments on MIMIC III

The Medical Information Mart for Intensive Care (MIMIC III) [22] database consists of electronic health records from patients in the ICU. We extracted a dataset with 1753 patients on antibiotics from MIMIC III. The patients' trajectories have 8 steps. For each patient, we have 26 patient covariates including lab tests and vital signs measured over time, as well as static patient features such as age and gender. The patient covariates change over time and are affected by the previous administration of antibiotics. Moreover, the treatment assignment mechanism is affected by the patient's covariates history and the previous administration of antibiotics [59, 60].

At each time step, we consider a binary treatment assignment problem, whether the patient should be administered antibiotics or not. Because the antibiotic treatment is decided daily for the patients, we use aggregate value for the time-varying covariates on each day since the ICU admission. We split the dataset into a training set (1000 patients) and a testing set (753 patients). We reuse the neural network hyperparameters and architecture that in the synthetic data experiments.

(a) White blood cell counts

(b) Treated v.s. Non-treated

Figure 4: Time-varying statistics of the training set.

A high white blood cell count is associated with severe illness and poor outcome for ICU patients [61]. Antibiotic administration in the ICU aims to reduce the white blood cell count. However, the effectiveness of the antibiotics treatment in reducing the white blood cell count is highly dependent on the time of antibiotic administration in a patient's covariates history. In Figure 4a, we report the mean and standard deviation of the white blood cell count over the patients in the training set. The

mean decreases in the first 4 time steps while it increases in the last 4 time steps. This shift happens even when the numbers of treated and non-treated patients have no significant change over the time steps, as shown in Figure 4b. This may be because antibiotics treatment become less effective if it is has been used in the previous time steps.

By DTR learning, we try to answer the question if using antibiotics treatment repeatedly is the appropriate treatment plan for decreasing the white blood cell count. We use our GRV-B algorithm to learn a DTR based on the training set. Because we cannot simulate the transition dynamics in this real-world dataset, we let the time $t$ treatment rule $\hat{d}_t$ in the learned DTR to make a treatment recommendation for patients in the testing set given their covariates and treatment history $\tilde{H}_t$. The testing set consists of the treatment trajectories of 753 patients. In Figure 5a, the majority of the patients would receive the treatments before the time step $t = 5$ in both the testing set (i.e. the observed DTR) and our DTR. This corresponds to the observation in Figure 4a that the average white blood cell count goes down in the first 4 time steps. The antibiotics treatment taken by the patients is probably the driving force for the decrease.

(a) Number of patients who receive treatments before the $t = 5$

(b) Number of treatments that the patients receive between $t = 5$ and $t = 8$

(c) Time gap of treatment assignment in the observed DTR and our DTR

Figure 5: Comparison of the observed treatments assigned in the testing set (observed DTR) and the treatments recommended by our DTR.

The treatment assignment of our DTR is different from the observed DTR from the time step $t = 5$ to $t = 8$. In Figure 5b, the number of patients who receive no treatments during this period of time is similar for the two DTRs. However, the observed DTR assigns almost 400 patients with a treatment at every time step between $t = 5$ and $t = 8$, while our DTR suggests most of the patients should receive 1 or 2 treatment over these time steps and there should be fewer patients who receive a larger number of treatments. The most striking observation from our DTR is that there is only a very small fraction of patients who need to receive four treatments in a row. In Figure 5c, the observed DTR assigns treatment much more frequently than our DTR. In Figure 4a, the white blood cell count goes up between $t = 5$ and $t = 8$, the treatment loses its effectiveness after being used many times. Our DTR suggests that we should decrease the number of treatments and enlarge the time gap between two subsequent antibiotics treatments, which aligns with the understanding in the clinical literature that the antibiotics treatment becomes less effective when it is used repeatedly.

## Footnotes

[2] https://github.com/xnie/adr