[Reviews · NeurIPS 2020]

Review 1

Summary and Contributions: The paper considers a framework for learning an optimal policy in a nonmarkovian dynamic treatment regime, which includes a regularization term that encourages the optimal solution (satisfying first-order necessary conditions) to solve the nonparametric estimating equation. The value functions are estimated by backwards recursion. Given that DTRs additionally take the history as input, a neural network architecture is imposed to use RNNs ---- Thanks to the authors for their response. I have read it. I agree the derivation is more difficult in the longitudinal setting due to the additional terms. I think, however, the previous use of this idea (algorithmically; and with multiple ablations) raises the bar a bit for this paper to provide evidence of the utility of the TMLE-inspired regularization. The paper is a bit short on this evidence beyond the numerical simulations (where again, it achieves good/admirable performance; but the experimental setup is not set up with enough ablations to isolate improvements from the RNN, NN architecture, or regularization term.) I think a more complete story here would really make this paper an accept. As it stands, this and other issues with explanation of the algorithm and architecture (main contributions) lead me to suggest a score of 6.

Strengths: methodological: Proposes gradient regularization to encourage the solution of a nonparametric estimating equation for a dynamic treatment regime. Proposes end-to-end learning framework with an RNN architecture to jointly learn outcome and propensity models Algorithms for learning optimal policies in DTRs are a bit lacking due to the difficulty of not only incorporating nuisance models for off-policy evaluation, but in re-estimating nuisance models in policy learning. (This is discussed in Zhang, Baqun, et al. "A robust method for estimating optimal treatment regimes." Biometrics 68.4 (2012): 1010-1018. and a preliminary approach is discussed there. ) While there is extensive work in estimation of dynamic treatment regimes, learning optimal policies with algorithmically sound techniques has been a weakness in the literature.

Weaknesses: The primary theoretical justification is given based on the first order necessary conditions satisfied by the regularizer term. The primary contribution beyond this is algorithmic (but some of the details of the algorithmic approach are a bit unclear). It seems to me that the appropriate baseline should be AIPW with a gradient based approach on some relaxation of the policy indicator functions (similar "outcome weighted learning" schemes appear in the biostatistics literature), see: Zhao, Ying-Qi, et al. "New statistical learning methods for estimating optimal dynamic treatment regimes." Journal of the American Statistical Association 110.510 (2015): 583-598. Given this, still, the paper can be viewed as applying targeted regularization to the task of policy learning. Certainly this leads to a large nonconvex problem but by appeal to the algorithmic success of neural networks, this can be viewed as an algorithmic contribution. Empirical evaluation: The empirical evaluation could be more structured since it provides the main justification for the approach. For example, results on policy improvement value could be primarily driven by using RNNS to capture temporal dynamics in the backwards-recursive outcome models.

Correctness: The claims are correct to the best of my knowledge. (There are some difficulties regarding the optimization landscape of neural nets and properties of local optima, but this is inevitable for the off-policy learning problem).

Clarity: The paper could be improved on the clarity front. The paper requires understanding different fields with extensive notation, so it is a difficult task. This could be accomplished by expanding the appendices to discuss some background to keep the discussion self-contained. (For example, expanding on the semi parametric efficiency discussion and derivation of the EIC could be helpful in the appendix). In particular, the presentation of some of the engineering decisions and architectures seems to suffer a bit due to lack of space. Given that a key contribution of the paper is the engineering of the overall policy learning system, it would be great to be more detailed in this regard.

Relation to Prior Work: The paper does a reasonable job situating itself relative to prior work given the breadth of relevant areas. However, a major concern is the relationship to the TMLE-inspired regularization approach taken in: Shi, Claudia, David Blei, and Victor Veitch. "Adapting neural networks for the estimation of treatment effects." Advances in Neural Information Processing Systems. 2019. I believe the fundamental observation re: the regularization term is the same as that paper (except this paper applies it for the more complicated EIC form for a nonmarkovian DTR, with a more complicated derivation). That is, that a regularization term can be added with the additional \epsilon parameter such that 0 partial derivative of that term implies \E_n EIC = 0. I may be a bit mistaken, but where exactly is the difference? Is the derived regularization a direct extension to the sequential setting, or is the proposed regularization in this setting a bit different? I went through the proofs but it is still not entirely clear to me where this differs; so further clarity/exposition on this front would be important. This would be important to get a clear sense of novelty and aid comparison to the batch setting. Another prior work that should be discussed is longitudinal TMLE for off policy evaluation in non-

Reproducibility: Yes

Additional Feedback: Certainly the paper is an ambitious project. The reason the score is not higher is because some steps are suboptimal (using a "softmax" relaxation of the indicator function rather than a properly calibrated scoring loss; refitting nuisance estimates for every policy value) and other steps are explained in a somewhat unclear fashion. Overall, the paper would benefit a great deal from additional ablation studies. It seems that the primary contribution is algorithmic, in addition to deriving the regularization term. (I find it somewhat unclear still why this should be expected to perform better finite-sample regularization). Comparing against ADR on the when-to-stop task seems a bit strange because contextual optimal stopping is such a specialized case. If one knew this were the task at hand, a more specialized approach on the structure of the value/q-functions would be favorable anyway. While it is certainly reassuring that good performance can be maintained even in this specialized setting (the general approach does not suffer too much relative to a specialized approach), it's hard to compare this approach to previous approaches in general DTR setting studied in the paper, such as optimizing the AIPW with the chosen optimization approach for the indicator function loss, eg lines 238-240 here. This would be a more direct/informative comparison. See: Zhao, Ying-Qi, et al. "New statistical learning methods for estimating optimal dynamic treatment regimes." Journal of the American Statistical Association 110.510 (2015): 583-598.


Review 2

Summary and Contributions: A crucial need in medicine is to decide how to optimally treat a patient. This problem can be defined in the context of a dynamic treatment regime (DTR), which is a sequence of individualized treatment rules based on the patient’s history and previous treatments. However, learning DTR based on offline data can lead to sub-optimal solutions since the bias due to time-varying confounders are ignored. This work presents a novel method called Gradient Regularized V-learning (GRV) to address this issue and learn the value function of a DTR.

Strengths: 1. The proposed GRV estimator is stable and achieves optimal asymptotic efficiency while relaxing the high variance caused by unstable inverse propensity score (IPS) product. 2. Theoretical results show that the optimality condition is met when the proposed regularizer is minimized. 3. Based on the designed estimator, two DTR learning algorithms has been developed. 4. Superior performance of the proposed method has been validated through both synthetic and real-world simulations. 5. Paper is well-written and fairly easy to follow.

Weaknesses: 1. According to the Table 1, ADR method shows competitive performance compared to the proposed GRV method. It is better to include a short discussion on their advantages vs disadvantages, e.g. their running time, stability etc. to better establish the advantage of the proposed GRV method.

Correctness: Yes

Clarity: Yes

Relation to Prior Work: Yes

Reproducibility: Yes

Additional Feedback:


Review 3

Summary and Contributions: They propose a gradient regularized V-learning to learn the optimal dynamic treatment regime. It is semiparametric efficient and allows people to use complex RNN.

Strengths: This estimator is statistically efficient. It allows the usage of complex deep learning methods.

Weaknesses: The paper is very complicated, and the explanation of the main algorithm is not enough. I could follow until Section 4.2. However, I feel puzzled in Section 4.3. For example, more explanation of the network architecture would be desired. More description of the objective function is also desired. The expression of the objective function is very long. The author has to give motivation why we should use it. Another thing is a comparison between DR. It says DR is efficient, but the finite sample property is bad because of the PS model. The proposed model also uses inverse PS and suffers from the same problem. Besides, the DR is much simpler to implement, and can partially avoid the problem by normalization of the inverse propensity score model. In this sense, I still cannot see why we have to use a very complex objective function in Section 4.3. Another thing the reader would want to know is the connection with the TMLE. As far as I know, TMLE is the way introducing epsilon, and updating the initial estimator so that the final estimator combining epsilon align with the efficient influence curve. It looks the proposed method does a similar thing. Is the proposed method fundamentally different from TMLE? >>>>>>>>>>>>>>>>>>.(After reading rebutal) The author tried to address my concerns. I updated the score. I understand the paper is super dense, and how to allocate the space is difficult. In my personal opinion, the most important part would be more explanation of the objective function. So I recommend the author to add it in the primary draft.

Correctness: I think so.

Clarity: The explanation and the introduction are clear. However, the main algorithm and the statement regarding the main algorithm (Section 4.3), which is the most important, is not well explained.

Relation to Prior Work: Well written.

Reproducibility: Yes

Additional Feedback:


Review 4

Summary and Contributions: Dynamic treatment regime (DTR) is a sequence of treatment rules indicating how to individualize treatments for a patient based on the previously assigned treatments and the evolving covariate history. The authors in this paper have proposed Gradient Regularized V-learning (GRV), a novel method for estimating the value function of a DTR. GRV regularizes the underlying outcome and propensity score models with respect to the optimality condition in semiparametric estimation theory and enables recurrent neural network models to estimate the value function under a DTR accurately.

Strengths: Dynamic decision-making in complex non-markovian environments are very challenging, especially when dealing with treatments of patients. It is crucial to give certain treatments in certain times in order to improve the conditions of the patients. This paper proposes a method to address this problem by formulating a value function and introducing an efficient method (GRV) to estimate the value functions using the recurrent neural networks (RNNs). The authors have also included theoretical analysis of optimality for their proposed method which adds to the value of the paper.

Weaknesses: There seems to me that there are not enough experiments in the paper. The authors have only included a simulated experiment in the paper and a real-world MIMIC experiment in the supplement. Even the MIMIC experiment is not very convincing for the practicality of the proposed method.

Correctness: They seem to be correct.

Clarity: Yes.

Relation to Prior Work: Yes.

Reproducibility: Yes

Additional Feedback: I liked the idea of the paper in using RNN for estimation of the value functions and also providing theoretical justifications for showing the solution of the proposed method. However, what cannot be seen in the paper is an actual real world problem which demands the existence of such a method. Although there is MIMIC experiment in the supplement, the results are not very convincing. For example, in Fig 6b the proposed DTR has suggested more patients to have 1, 2, and 3 treatments between the time steps t=5 and t=8, and it has only suggested fewer patients with #treatment=4. So, it is hard to interpret the results and claim that the proposed DTR outperforms the others. I think some other convincing examples could add to the value of the paper and the practicality of the method. The response has been read.

[Author Response · NeurIPS 2020]

We thank all reviewers for their thorough assessment of our paper and suggestions for improvement.

**Response to reviewer #1**

• ***Experiments, clarity and additional feedback.*** Thank you for your insightful feedback. We agree with you and in the revised paper, we will add an ablation study of RNNs and GRV, implement the combination of benchmarks you suggested, and add more simulation studies to provide a more complete evaluation of our method. We note though that RNNs are not necessarily superior, because the models in other methods can also capture the data generating functions. Finite sample regularization improves the estimation because it adapts all the initial models to solve the nonparametric estimating equation (EE), rather than after the initial models are trained and potentially overfits to the training data. Note that validating causal models is tricky because we do not observe the counterfactuals. We will definitely improve the paper clarity in the revision. As you suggested, we will expand the Appendix to provide a notation table, add several sections introducing the needed background from the various fields such that the paper becomes more accessible and our derivations are easier to follow. Thank you for the excellent suggestions.

• ***Relation to prior work.*** Both GRV and Targeted Regularization (TR) [Shi, et al.] are motivated by solving the EE. However, note that the EIC of DTR in GRV is much more complicated than the EIC of ATE in TR. The EIC of ATE only contains two outcome models and one propensity score model. The EIC of a DTR has many terms as shown in Equation (6-8). This is because the value function estimation involves the DTR, the outcomes, and the treatment assignment mechanism over time. We need to solve the EE by adapting a sequence of outcome and propensity score models over time. There is no direct implementation of TR which can solve the EE across time steps jointly in the DTR learning setting. The GRV regularizer is designed to simultaneously solve the complex EE of DTR learning, which involves models at different time steps; this requires a more technical derivation than TR. In addition, re-estimating nuisance models in DTR learning is more challenging than in ATE estimation. The current related work section gives an overview of the estimator classes and relegates the discussion of the longitudinal TMLE for OPE to the Appendix. In the revision, we will make the connection with TR and TMLE clear in the main body of the paper. Thank you.

**Response to reviewer #2**

Thank you for the useful suggestions. We agree and we will expand the Appendix to include a discussion of advantages and disadvantages as compared to the other methods, including the running time, stability etc. Thank you.

**Response to reviewer #3**

• ***Motivations and clarity.*** Thank you for your comments. We agree with you and we will improve the presentation of our method and the adopted architecture in Section 4.3. Due to space limitations, we already relegated some of the discussion about the network architecture to the Appendix. The objective function in Equation (9) has two terms, $\mathcal{L}_t$ and $\mathcal{R}_t$. The term $\mathcal{L}_t$ includes the standard loss function for the outcome regression models and propensity score models in causal inference. In DTR learning, we also need sequential regression to estimate the long-term outcomes; this is why we have the last term in $\mathcal{L}_t$. The regularizer $\mathcal{R}_t$ is motivated by Theorem 1. Minimizing $\mathcal{R}_t$ encourages the fluctuated outcome models to solve the optimal score equation i.e. nonparametric estimating equation, on the R.H.S of Equation (10), so that the resulting estimator is efficient for estimating the value function. In the revised manuscript, we will improve the presentation in Section 4.3 and explain clearly the motivation of each term in our objective function.

• ***Doubly robust (DR) and TMLE.*** The DR estimator achieves efficiency by applying the DR estimating functional on some pre-trained initial outcome and propensity score models. An example of DR estimator is given in Equation (4), where the inverse propensity score (IPS) is multiplied by the cumulated outcome and $Q$ functions. In GRV, the initial estimators are optimized to solve the nonparametric estimating equation jointly with an additional parameter epsilon; this provides additional flexibility to solve the estimating equation. Our estimator in Equation (11) is stable because the IPS is multiplied by a small epsilon. Similarly, it is well known that solving the estimating equation using an additional epsilon in TLME can give a more robust estimator than DR in ATE estimation. The parameter epsilon enables trading-off between finite sample stability and asymptotic efficiency. However, TMLE is not robust when the initial models are not well specified or estimated. This drawback exists even in the simplest setting of ATE estimation and becomes even more severe in DTR learning which involves many outcome and propensity score models. Our method adapts all the initial estimators and the epsilon to solve the estimating equation jointly by regularizing the models in training. By reducing the dependence on the quality of the initial models, our method is thus more robust than TMLE for DTR learning. The connection between our method and TMLE is discussed above Section 4.3 and in the Appendix. We will further improve the discussion and make these points clear in the revision. Thank you.

**Response to reviewer #4**

Thank you for your useful feedback. In the submitted paper, we have included two simulation studies to evaluate the method performance (in terms of both DTR evaluation and optimization) across various sample sizes. We note that, unfortunately, it is impossible to use offline datasets to benchmark the methods; existing DTR or policy learning papers are also based on simulation studies. We use the MIMIC III dataset for illustration only, to show that the learnt DTR will lead to gradually decreasing the treatment, which corresponds to the clinical literature. The real DTR application can only be demonstrated well through real-world experiments and simulation. In a real-world test-bed, the clinician can check if his/her treatment decision agrees with the DTR before prescribing it. In the revision, we will also explain how such test-beds can be built to evaluate DTR. Thank you.

[Meta-Review · NeurIPS 2020]

The reviewers found the gradient regularized V-learning algorithm proposed in this paper to be novel and to address an important problem in the application of machine learning to clinical applications. They pointed out a number of advantages including the theoretical properties of the approach, the ability to incorporate neural networks into the V-learning framework, and the performance of the algorithm on the data in the experiments. There were a few conerns about the paper, most of which having to do with the experimental evaluation, and others the relationship between the proposed method and some existing work. In particular, the reviewers thought that there were other baselines that would be more appropriate to compare with and that there wasn't a "real experiment". Additionally, some reviewers found the paper to be very complicated and unclear in some places. The authors provided a detailed set of feedback that addressed the main criticism of the reviewers. In the discussion phase the most reviewers indicated that the authors had addressed their concerns in a reasonable manner. In one case, a reviewer lowered their score a bit and pointed out the need for principled ablation studies as in the paper by Shi, et al. Despite this decreased score the reviewers still came to the consensus that the paper should be accepted. This is a dense paper and the reviewers offered some helpful feedback to improve the clarity of the paper. Additionally, there are many experiments that can be included in an expanded version of the paper. As written, this is a good NeurIPS contribution.